# Agent-Based Simulation of Hardware-Intensive Design Teams Using the Function–Behavior–Structure Framework

**Mitch Bott [1],***  **and Bryan Mesmer [2]**

[1] Computer Science, University of Alabama in Huntsville, Huntsville, 35899 AL, USA
[2] Industrial & Systems Engineering and Engineering Management, University of Alabama in Huntsville, Huntsville, 35899 AL, USA
* Correspondence: mjb0021@uah.edu; Tel.: +1-801-620-0917

**Abstract:** Agile processes have been used in software development, with many case studies indicating positive changes in productivity when these processes are used. Agile processes are beginning to be applied to work beyond software-centric systems. There does not yet exist a diverse set of studies on the effectiveness of Agile processes on hardware-intensive systems. The research in this article applies a modeling and simulation-based approach which uses the function–behavior–structure framework to evaluate the effectiveness of waterfall and Agile processes. The simulation was validated against case studies of software-centric design efforts. When applied to a space launch vehicle—a highly coupled, hardware-intensive system—the simulation shows that the benefits of Agile may not be as great as those seen with software-intensive systems.

**Keywords:** agent-based model; Agile; design; waterfall

## 1. Introduction

A challenge in systems engineering is the evaluation of the effectiveness of different engineering processes. The complex system projects that engineers and designers work on are often large, with many individuals involved, span multiple years of development, and only use a single process. Measuring the effectiveness of different processes with the same team developing the same system is cost prohibitive for large and complex systems. This study examines agent-based simulations as a way to explore the effectiveness of different engineering processes [1–3], focusing on Agile and waterfall processes.

Agile processes are often used in software development and have begun to be used in non-software fields. These processes are based on the Agile Manifesto [4] and favor small, self-organizing teams working together to create a solution. This study examines the effectiveness of Agile methods on large, hardware-intensive, complex, and coupled systems. First, an established agent-based simulation of design teams was used to simulate waterfall and Agile design practices for a large software program to demonstrate the validity of the simulation approach. The simulation was then used to determine the effectiveness of waterfall versus Agile processes on a space launch vehicle—a large, hardware-intensive, complex, and coupled system.

The detailed results of the research leading to the simulation development are presented in Section 2. Section 3 presents a discussion on the findings of the research. Section 4 concludes the paper, highlighting the new contributions of the paper in the modeling and simulation field, as well as the contributions in systems engineering, to better understand the use of Agile processes on large and complex systems.

## 2. Results

The results consist of a literature review of background and supporting material in Section 2.1, an overview of the methods used to develop the estimate in Section 2.2, a review of the design of the simulation in Section 2.3, and an overview of the verification and validation efforts of the simulation in Section 2.4.

### 2.1. Literature Review

#### 2.1.1. Agent-Based Modeling and Simulation

Agent-based models use "agents"—autonomous entities—to represent some system of interest [2]. Agent-based models are typically deployed into a simulation where they interact with other agents, are directed to meet a goal, and are given the ability to learn and adapt [3]. The agents in this study are not the system being developed, but rather the system of designers and engineers that are developing the system.

Agent-based models can be used in simulations of complex phenomena [3]. Since large design teams create complex behavior [5], where the behavior of the whole is more than the sum of its parts, an agent-based approach to modeling the problem can be useful to expose the complex behavior. In this study, the agents perform design tasks using either Agile or waterfall methods. Agent-based models require a reasonable level of abstraction to credibly model most problems [3]. This necessitates a somewhat simple model of an agent. In this research, agents are individuals performing design tasks, and they are represented with models that simplify the cognitive process they use to develop a design, as discussed in Section 2.1.2.

#### 2.1.2. Models of Designers

Many different types of models of designers and designing have been created, including ones that base the model on negotiation between designers [6], developing design values for optimization [7], data flow between designers [8,9], and moving between various states of design [10]. In this research, a well-validated and simple model of designers was chosen for the agents to focus the research on the simulation aspect rather than developing a new model. A model of designers changing states during the design process, the function–behavior–structure (FBS) model, was chosen.

Gero [11] created the FBS model as a way to standardize design ontology. The FBS model contains states and transitions. The states describe a part of the design process that a designer is in, and the transitions describe the activity the designer needs to complete to move to another state. The ontology is simple and generic, allowing the FBS model to be a general-purpose model that can span multiple types of design efforts.

The FBS model first emerged with some aspects between states not fully defined [12]. A corrected FBS model was later presented [13]. The FBS model is shown in Figure 1. The states of the model are requirements (R), expected behavior (Be), structure (S), behavior of the structure (Bs), and description (D) [10,11,14]. The transitions between states are formulation, synthesis, analysis, evaluation (which does not actually result in a change in state), documentation, and three different types of reformulation [15].

The FBS model has been validated through verbal protocol analysis [10,16–19]. These studies have shown that FBS can be used to represent the various states that a designer is in while he or she is performing design work [10,16–19].

The FBS model provides an abstracted model of a designer that can be adapted to an agent model. It has been through validation studies and is general-purpose in nature, allowing it to be used to study a variety of design efforts.

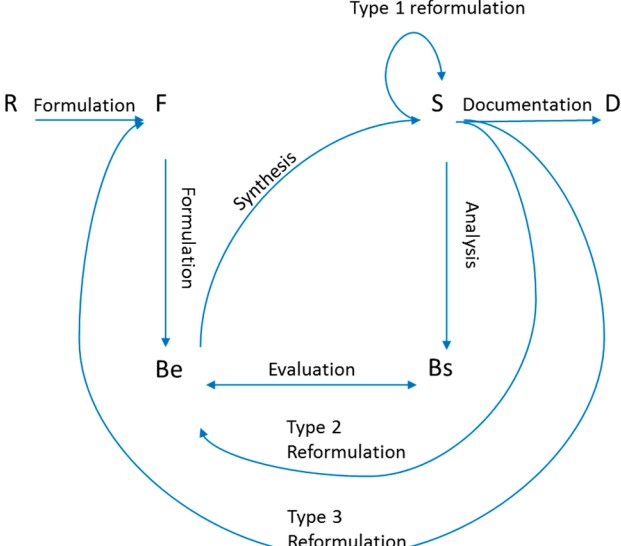

**Figure 1.** The function–behavior–structure (FBS) model.

### 2.1.3. Design Synthesis Process

Design synthesis processes were used in this research to drive the agents through their state transitions and govern the simulation. These processes often synthesize a design through a hierarchical breakdown of the design, which then creates elements small enough that an individual can design it [20,21]. This type of process was used in the simulation to govern the behavior of the design agents and create pseudo-designs.

Generic design processes exist in standards such as ISO/IEC/IEEE-15288 and MIL-STD-499 [22,23]. These standards allow significant tailoring, as it has been found that a rigorously defined design process cannot handle all possible design problems [24].

Waterfall is one process used for design synthesis. It involves a sequential process where designers are broken out into teams where each team works on their portion of the design through the design process. This work is synchronized for the system at major review checkpoints and the design matures, as a whole, in a set of successive events [25]. The waterfall process works well when the scope of the design has low uncertainty, such as when a minor change to an existing product is being performed [26]. This results from waterfall doing all planning up front and assuming perfect or near-perfect knowledge about how the design effort will be executed [27]. The waterfall process is not often advocated for new projects, but it remains a mainstay of many large and complex system developers [28].

Agile and waterfall design synthesis processes for software development have been analyzed in system dynamics models [29]. These models analyzed the dynamics of the process and their effect on error rate and time taken to complete work, showing that waterfall takes about 80% longer to complete a moderately sized software project [29].

### 2.1.4. Agile Processes

Agile development began after the Agile Manifesto, a document outlining the principles behind Agile software development, was published in 2001 [4,27,30]. Agile was originally envisioned as a methodology for better software development. The main principle of agility outlined in the Agile Manifesto is realized by empowering software developers and relying on their technical excellence to create simple designs to meet customer needs [31]. This agility allows designers to respond to changes in requirements and business needs easily.

Agile processes are used for design synthesis. They have been extended for use on all types of systems, not just software [32,33]. Most Agile design process examples are focused on software [34], while some Agile processes have been created for hardware-centric systems [35,36]. Agile processes tend

to focus on the ability to rapidly respond to change, as documented in the Agile Manifesto [4,27,30]. Agile processes also focus on the use of the abilities of the developers to create solutions for customers [31]. In this study, Agile processes were simulated to examine their benefits and drawbacks when compared to a more traditional waterfall approach. Specifically, the Agile process of Scrum was examined.

Scrum is by far the most widely used Agile process, being used about 5 times more frequently than other Agile methods [37]. Scrum claims to result in teams performing better than the sum of each individual member's abilities [38]. Claims of the benefits of adopting Agile processes for software development tend to vary greatly, with most teams reporting productivity increases between 36% to 42% and extreme cases reporting project failure or as much as a sixfold increase in productivity [30,39]. This study tested these claims through simulation.

Waterfall and Agile design synthesis processes were both used in the agent-based design simulation. Published examples of these processes were used in designing and structuring the simulation [22,30].

### 2.2. Methods

This paper uses agent-based modeling and simulation to examine the difference in productivity when teams use waterfall or Agile processes. The research was started with a literature review of relevant topics. The literature reviewed helped to inform the approach to develop an agent-based simulation of designers. Agent-based simulation is a common methodology used to examine complex problems [2,3]. Due to the size, number of variables, and expense of testing, systematically testing the difference in team productivity with human experiments when using Agile versus waterfall is not feasible.

First, existing models of designers were examined, and an existing, validated model was selected. The model was then calibrated with real-world data. A simulation of the development of a software system was created using the calibrated model. This simulation can easily be validated against published data, as the study of software development when using Agile is extensive. The same methodology was used to examine a hardware-intensive system: a launch vehicle. The results from this simulation were used to examine the effectivity of Agile processes when used on hardware-intensive systems.

This work builds on the work of Gero and others into the modeling of designers to, in this paper, examine engineering organizations and the systems engineering approaches they adopt. The simulations are novel, and we demonstrate that the methodology, when applied to a software system, yields valid results. The methodology also yields new insights into the nuances of using Agile methods on hardware-intensive systems.

### 2.3. Modeling and Simulation of Design Teams Performing Agile and Waterfall Development

This section outlines the work done to develop an agent-based model and simulation of design teams performing waterfall and Agile work on two example systems: a software program and a launch vehicle. The section details the model used for the agents and how the simulation utilizes waterfall and Agile concepts. It reviews how the model was calibrated for use in this application. The details of both the software program design simulation and the launch vehicle design simulation are presented.

The primary difference between the launch vehicle and software design simulations is the modeling of the coupling in the design of the launch vehicle. Coupling is the implicit dependency of two pieces of the system on one another [40]. The software design simulation was built with the assumption that each subsystem design team could be built in isolation from the others with internal subsystem characteristics not affecting other subsystems. This was not assumed to be true for the launch vehicle, where decisions for each subsystem have an effect on the system performance.

#### 2.3.1. The Agent Model

Designers were modeled as agents using the FBS model described in Section 2.1.2. The FBS model was mechanized as a first-order Markov process where transition from one state to the next is governed by a probability. This allows the FBS model to be simulated as a probabilistic process. The Bs and

S states of the FBS model were collapsed into a single state for the purposes of modeling, which is represented by the dashed box in Figure 2. This collapses the synthesis, analysis, and evaluation activities into a single activity. This was done so that the FBS model could be represented as a Markov process. As is, the FBS model could not be represented as a Markov process since the path from F to Bs has no return path. Also, the evaluation process does not actually result in changing states. This may cause the FBS model to lose some amount of fidelity, but it still allows the overall FBS flow to be represented so long as the analysis and evaluation activities are captured as part of a modified synthesis activity. An example of the model with notional transition probabilities is shown in Figure 2.

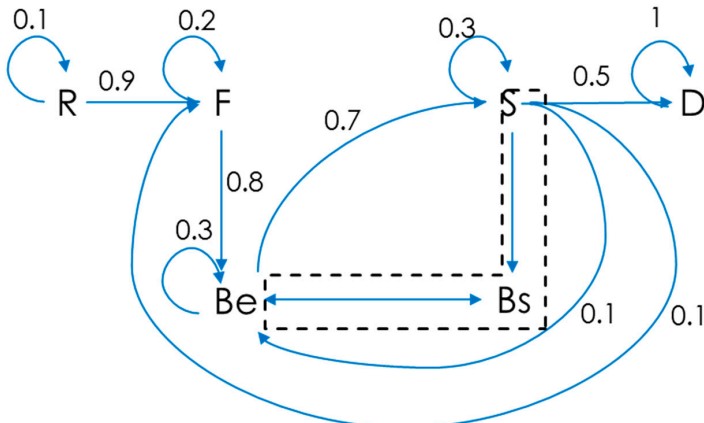

**Figure 2.** FBS model setup for Markov simulation with example state transition probabilities.

Agents were represented in the simulations with the FBS Markov process. The model keeps track of the agent's current state. Each of the agents was simulated as transitioning through the implemented FBS model using random draws to calculate a value that was compared to the transition probabilities to determine the state for the next time step.

The agents were arranged in a hierarchical structure representing the teams that build the various parts of the system. In the case of the software system development, the computer program had four major modules. Each module comprised two to four sub-modules that were developed by individual teams. This is represented in Figure 3, with each development team represented by a box in the figure.

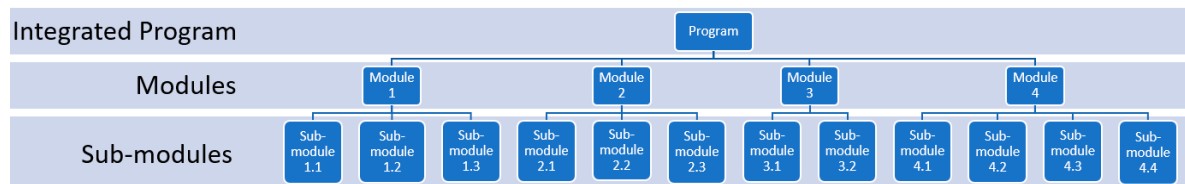

**Figure 3.** Representation of the team and program structure for the software program design simulation.

At the integrated program and module level, only one agent was used to represent the integrating functions. These agents are equivalent to team leads in the waterfall construct, or product owners in the Agile construct. The teams at the sub-module level each consist of eight designers. Eight was chosen as the team size to match scrum recommendations [30]. The sub-modules were further divided into ten individual functions or features that the design teams create at the sub-module level.

For the launch vehicle design simulation, the agents were arranged in a hierarchical structure representing the teams that build the various parts of the launch vehicle. There was an intermediate-level scrum team for Stage 1 with scrums developing the engine, propellant tanks, structure, and defining requirements. Another intermediate-level scrum was for the second stage with similar scrums as the first stage. There was also a third intermediate-level scrum that defined the payload and mission profile with a single scrum. Each of the scrums had a lead that participated in planning and performing

the design work. There was also an overall lead for the design effort. Each of these individuals was represented as an agent. Each sub-team had eight members based on recommendations for the size of scrum teams [32]. This organization is shown in Figure 4 below with the groups in the Scrum of Scrums shown horizontally across the organization. The same structure was used in the waterfall simulation as well. The top scrum group, along with the intermediate scrum group, set the initial goals for the project and monitor progress. The lowest "Scrums" group is where actual design work is performed. In the scrum simulation, the intermediate and top scrums had minimal participation in the design work, per scrum guidance [32]. In the waterfall simulation, the intermediate and top scrums were more involved and participated in the lower-level design work.

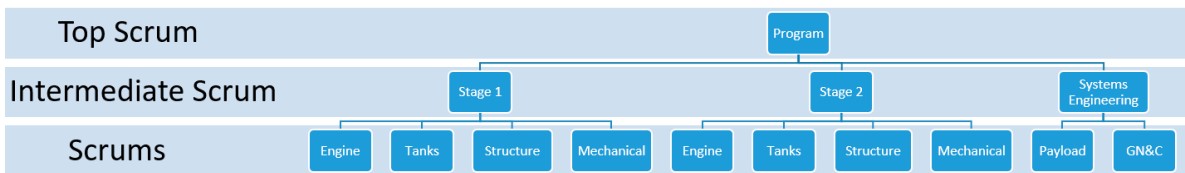

**Figure 4.** Launch vehicle design organization.

### 2.3.2. Waterfall Simulation

The waterfall simulation was created to follow a typical development program through several milestones. These are typical design milestones and are advocated for in traditional systems engineering standards such as MIL-STD-499 [23]:

- System requirements review (SRR);
- System functional review (SFR);
- Preliminary design review (PDR);
- Critical design review (CDR).

For the waterfall simulation, these design phases were mapped to the FBS model as follows:

- SRR—corresponds to the transition from R to F

    ○ A goal of SRR is to translate customer requirements into system-specific functions [23]. This aligns well with the FBS transition of formulation where functions are formulated from requirements.

- SFR—corresponds to the transition from F to Be

    ○ A goal of SFR is to create a design approach that performs in such a way as to accomplish required functions [23]. This aligns well with the FBS transition of creating expected behavior from functions.

- PDR—corresponds to the transition from Be to S

    ○ PDR is meant to show that the detailed design approach for the system satisfies functions [23]. The Be to S transition is meant to achieve this as the expected behavior (low-level functions) is used to derive the design.

- CDR—corresponds to the transition from S to D

    ○ CDR is meant to show that the total system design is complete, meets requirements, and is ready to be built or coded [23]. This is represented by the transition from structure to documentation. In order to make this transition, the design must be complete, which is met by completing the structure phase; it must meet requirements, which is met by

completing the Bs to Be comparison (part of the structure phase in the implemented FBS model); and the design must be ready to be created, which is represented by completing the documentation, which is where the design is handed off to manufacturing or coders to be created.

All agents start the simulation in the requirements state, R. The work to get to SRR begins with a hierarchal flow down of requirements. The agents are arranged into three levels of scrums, as described in Section 2.3.1. The engineer at the top level provides information to engineers at the next level down. At this middle level, the information is further refined into lower-level system functions, which are then provided to the teams at the lowest level to develop functions from. The work to get to SFR (or transition to Be) follows this same hierarchal construct as the agents go from F to Be. The work to get to PDR only consists of the lowest-level teams transitioning to S, as they are the ones that develop the design. The final step involves the lowest-level agents transitioning to D, which is also where rework may occur since the behavior of the structure of the design is compared with expected behavior and reformulations may occur. The waterfall simulation process is illustrated in Figure 5, where the FBS state transition that corresponds to the process phase the designer is in is highlighted in the FBS model. After all agents have transitioned to the documentation phase, the simulation ends.

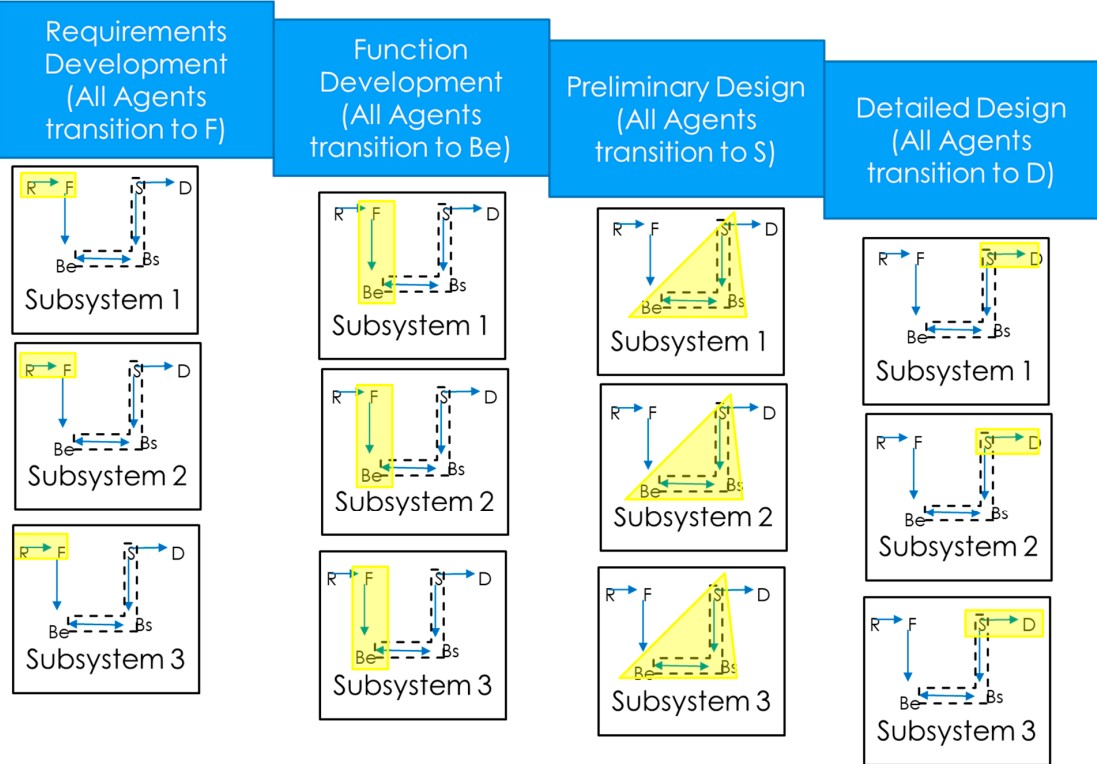

**Figure 5.** Notional waterfall design process.

### 2.3.3. Agile Simulation

In the Agile simulation, the agents move through the FBS states per the scrum process. The top- and middle-level agents begin the process with a planning phase. To represent this planning phase, these agents must transition from the R state to the F state, representing the work to develop high-level requirements into functions and the initial population of the work backlog. This represents the typical work done to develop user stories. User stories are a method used in Agile to develop system requirements [41]. From here the work is passed directly to the lower-level teams to perform sprints.

Each of the teams at the lowest level performs sprints to develop functions within their subsystems. The goals of each sprint are to accomplish the work needed to move to the next state in the FBS diagram.

Reformulations represent rework on the product that is performed according to the product backlog. The product backlog can be populated with fixes to the structure, behavior, or function depending on the type of reformulation.

Figure 6 outlines the scrum process and its ties to the agents in the simulation. The process starts with an initial planning phase where the top two tiers of agents must transition from R to F. After this the lower-level teams are set to work through sprints to accomplish their work. They go through sprints that can notionally last from 10 to 30 days with a daily scrum. At the end of the sprint, a product iteration is produced. The product backlog is then consulted to determine what features/fixes need to be put in the product next. After all agents have transitioned to state D, the simulation ends as the documentation of the design has been completed.

Figure 7 shows the notional Agile process which starts with key agents going through initial system definition by going from the R to the F state. Yellow highlights indicate the FBS state transition that corresponds to the step in the process that the designer is in. Lower-level agents then develop functions in a serial manner by moving through all the FBS states.

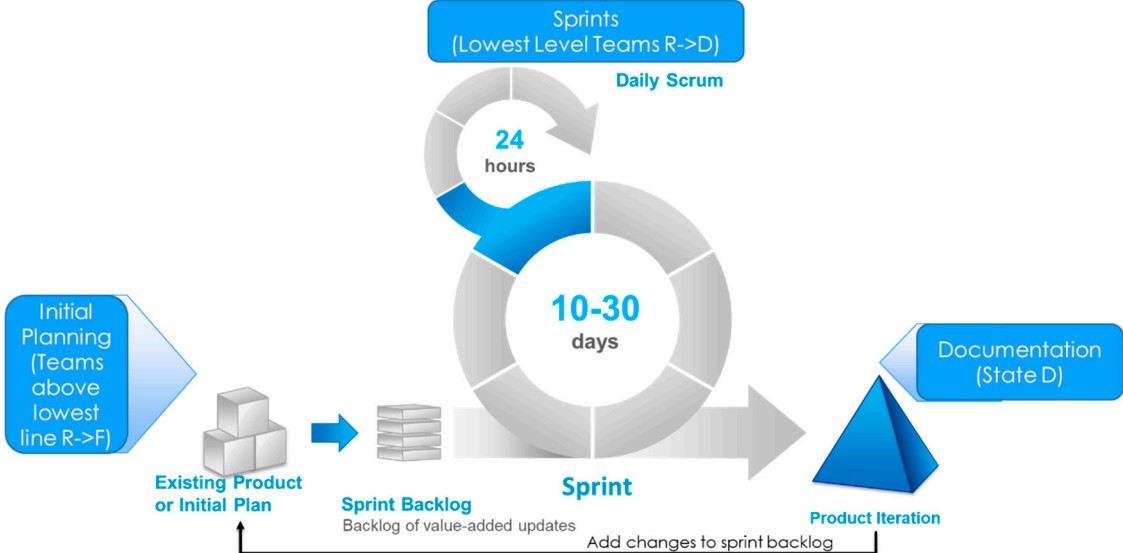

**Figure 6.** Agile process overview.

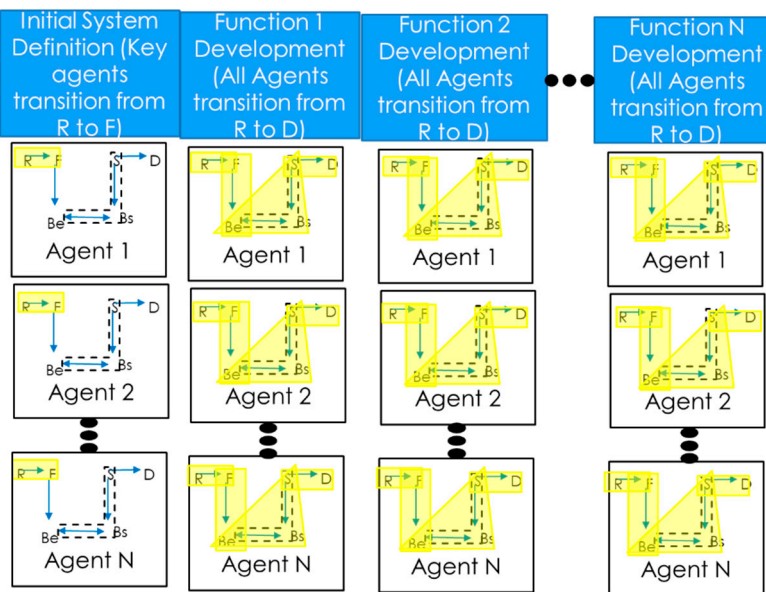

**Figure 7.** Notional Agile design process.

### 2.3.4. FBS Model Calibration

The FBS model requires transition probabilities for it to be used in a simulation of how designers work on real problems [17]. Existing data from FBS validation efforts are based on mock design sessions for simple systems [14,17]. For this research, data from the development of a complex aerospace system were used to calibrate the FBS model and derive transition probabilities [42].

### 2.3.5. Simulation Design

The simulation was developed in the MATLAB® programing language. It was chosen due to the familiarity the author has with it and its inherent ease of analyzing output data and plot generation. The language includes a large library of functions that are useful for Monte Carlo simulations, such as the multiplicative lagged Fibonacci generator [43] used for pseudo-random number generation. See the Supplementary Materials section for a link to the source code of the simulation and outputs of the simulation.

The agent model was implemented as a $5 \times 5$ array which tracks the transition probabilities between the FBS states. The mean FBS performer and an array of the difference between the mean performer and the high side of the 90% confidence interval on transition times were used with the pseudo-random number generator to create the individual agent models. This resulted in multiple arrays of FBS transition probabilities that ranged between the lowest and highest possible performing agents.

The agents were assigned to teams in the hierarchy structure shown in Figure 3 for the software simulation or Figure 4 for the launch vehicle simulation. Certain agents were assigned to be team leads, the overall program lead, or individual designers. These agents maintained their position in both the waterfall and Agile simulations, simulating the same team being used for each process.

Waterfall Simulation Design

In the waterfall simulation, all the agents perform the following steps in a sequential fashion (FBS state transition shown in parentheses):

- Requirements development (R→F);
- Function development (F→ Be);
- Preliminary design (Be→S);
- Detailed design (S→D).

To step through each of these steps, a random draw is used with the transition probabilities in the agent model to determine if the agent advances to the next state in the FBS matrix. If the random draw is greater than the transition probability, this indicates that the agent has completed the activity needed to transition to the next state. If the random draw is less than the transition probability, the agent remains in the state it is in until simulation time advances and a new transition probability is created to check if the transition has occurred.

In order to move from one step to the next, all agents have to complete the corresponding state transition for all functions they are responsible for. After each time step advancement, agents that have not yet transitioned to the next state are tested to see if they advance with a new random draw. Once all agents have advanced, the simulation gathers metrics about the current step the design is at. These include total effort hours expended by all agents advancing the design, total time needed to complete the step, and for the critical design phase, total time spent in rework. After all steps are complete, these metrics are compiled into metrics for the entire simulation.

The simulation was repeated 10,000 times as a Monte Carlo simulation [44]. Data from all the simulations were compiled together to analyze the performance of the design team using waterfall processes. The average time needed to complete the design, average number of effort hours needed to complete the design, and average amount of time spent in rework were calculated from the compiled simulation metrics.

Agile Simulation Design

The Agile simulation starts with a planning phase where the program leader and team leaders perform initial requirements development going from state R to F. The agents advance through the FBS states in the same manner as that used in the waterfall simulation.

Next, each of the design teams (lowest level of the organizational hierarchy) performs Agile sprints. They move through the entire FBS model, starting at state R and ending at state D for each of the design functions they are responsible for in a serial manner. The teams work in parallel to one another without the need to synchronize work between them at major reviews, unlike the waterfall team. Instead, when functions are ready, they are added to the product incrementally and released, eventually building a complete product.

Like the waterfall simulation, the Agile simulation was repeated 10,000 times as a Monte Carlo simulation. The same performance metrics were gathered during each iteration and compiled into metrics that span the 10,000 iterations. The averages of these metrics were also calculated.

After the Agile simulation completed, the waterfall and Agile simulation metrics were compared to each other. This was done by determining the ratio of average time to complete the design, ratio of average effort to complete the design, and ratio of time spent in rework.

2.3.6. Assumptions

Every model and simulation consist of assumptions. The model and simulations used in this research, along with all others, are approximations of reality [45]. This section consists of a list, in no particular order, of simplifying assumptions used in the software program and launch vehicle design simulations. The reasons for these assumptions are also listed. These assumptions were needed to make this particular effort tractable, as representing all aspects of a large team of individuals developing a complex design in simulation is viewed as a nearly impossible task.

- Designers can be represented by the FBS model. The FBS model is an abstracted model of designers. While it does not represent all aspects of designers, it contains enough information to represent the states that the designers go through during the design process [17].
- The synthesis, analysis, and evaluation activities were combined into a single activity. This was necessary so that the FBS model could be represented by a first-order Markov process.
- The system being designed is unprecedented. The designers do not know a priori the optimal design solution.
- Design teams work in parallel. Teams do not wait for other teams to perform their work before starting.
- Reformulations caused by design incompatibility are type I reformulations. It was assumed that these types of reformulations are caused by incompatibility in the structure of two different parts of the design.
- The simulation was designed assuming that the agents interact with one another through a model of the system they are developing. Communication between the agents was not modeled due to this assumption. Rather, agents learn of the design decisions and implications of those decisions on the system as soon as their peer agents learn them. Thus, information learned about the design of the system is given to agents with zero time lag.
- Team leaders do not contribute to design work. This was done to represent the roles of these leaders primarily in the planning of the design through requirements and function derivation.
- Idle time was not modeled. It was assumed that agents that complete their work early have other projects they can work on and their idle time does not count towards the total number of effort hours needed to complete the design.
- Agents understand the coupling in a design. When coupling forces redesigns of subsystems, the minimum number of subsystems are redesigned.

- The software and launch vehicle systems are simple versions of these types of systems. This assumption was needed to ensure that the simulation development effort was tractable as a fully defined development process for these large and complex systems which are difficult to define and fully simulate. This makes the simulation results not necessarily representative of real-world performance outright. They are still considered valid for comparison purposes, which is the primary objective of this research.

### 2.3.7. Software Program Development Simulation

The software design simulation was created following the design outline in Section 2.3.5. Agents were arranged in teams as per the hierarchy shown in Figure 3. Each team was assigned to develop a single module of a software program. The exact purpose of the software is not necessary to model so long as the complexity of the work for the software team is properly modeled.

Each of the agent teams has eight members. The teams each have a software sub-module to develop with equal complexity to one another. Each software sub-module has 10 functions that require requirements development, design, and evaluation. This means that designs must visit all FBS states for each function to adequately design it. In total, the software program has 120 functions that need to be developed. The total number of individuals represented by the simulation is 101.

The sub-modules were assumed to have a low amount of coupling [46], and the design of one submodule does not affect other submodules. Developing computer programs with a low amount of coupling is standard design practice for software systems [40]. This is not true for the launch vehicle development simulation.

The simulation performed 10,000 Monte Carlo iterations each of the waterfall and Agile design approaches. These simulations are described in Section 2.3.5. At the end of each of the simulations, metrics were gathered and tallied. The total time needed to develop the design, the total number of effort hours expended to develop the design, and the time spent doing rework were all calculated for each of the simulations. The ratio of the average of these metrics between the Agile and waterfall simulations was also calculated as a way to compare the two approaches.

### 2.3.8. Launch Vehicle Development Simulation

The launch vehicle design simulation was built using the same simulation structure and techniques as the software program development simulation, but it contained an aspect that made it more complex than the software program development simulation. This is the coupling of design choices between design teams. To simulate this, design variables were added to the simulation to represent the design choices of the agents. The choices of these variables can introduce the need to reformulate a design that is working fine by itself but does not integrate with the rest of the launch vehicle. A simulation of the ascent of a launch vehicle was created to determine what combinations of design variables resulted in a successful design.

The same random draws that govern the agents transitioning through the FBS model were used to pick design variables related to the launch vehicle. As part of the evaluation step of the FBS model, the design structure was compared, not only to its expected behavior, but in terms of how it impacts the expected behavior of the system given the design choices from the other design teams. This provides some insight into how a highly coupled design can impact the work of design teams. Figure 8 shows two design variables represented as two random variables. The intersection of the two variables that creates a valid design is shown in blue as the "D" space. Type I reformulations occur in the space outside of the valid design space and require rework to reformulate the design to the point that it could be a valid design. In the launch vehicle design simulation, a total of nine design variables was needed to align to a space where a design was valid. The nine variables were chosen since these are the variables needed to define the launch vehicle for a 3-degree-of-freedom simulation to determine its capabilities.

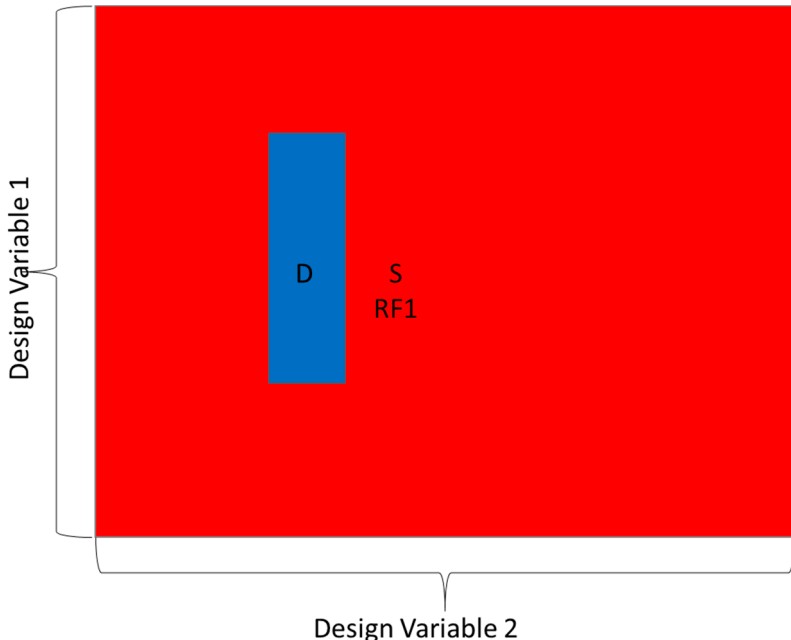

**Figure 8.** Illustration of how design variables are tied to FBS states.

Nine design variables in total were used to design the launch vehicle. They belong to different engineering disciplines represented by different agents in the FBS model:

- Stage 1 thrust—determined by the Stage 1 rocket engine team;
- Stage 1 propellant mass—determined by the Stage 1 tank team;
- Stage 1 structure mass—determined by the Stage 1 structural team;
- Stage 1 diameter—determined by the Stage 1 mechanical team;
- Stage 2 thrust—determined by the Stage 2 rocket engine team;
- Stage 2 propellant mass—determined by the Stage 2 tank team;
- Stage 2 structure mass—determined by the Stage 2 structural team;
- Stage 2 diameter—determined by the Stage 2 mechanical team;
- Payload mass—determined by the systems analysis team.

The launch vehicle simulation concluded with the nine design teams choosing design variables that, perhaps after many reformulations, result in a design that successfully injects a payload into orbit. Metrics were gathered by the simulation to determine how long it took to develop the design, how many effort hours were expended, and how much time was spent doing reformulations.

*2.4. Verification and Validation*

The simulations were verified through several steps. The code was debugged using the MATLAB® built-in debugger. The simulations were verified to be executing properly by using a simplified FBS transition matrix. The matrix was set to equal-probability transitions between states, and then the state transition times were examined. The first passage time on average should be the reciprocal of the probability of transition [47]. For example, in one case, the probability of going from R to F was set at 0.5. With 100 samples run, the average first passage time for the 10 agents at the top and middle levels was 2.051—very close to the theoretical answer of 2.0. Further checks of the simulation were performed to verify proper operation, including checking other first passage times with the notional FBS transition matrix and checks of the routines used to sum up the final statistics of the program. The number of Monte Carlo runs was examined with 100 and 10,000 runs being performed with both simulations. The results from the 100-run simulation were found to be within 5% of those from the

10,000-run simulation. This provided evidence that 10,000 runs were sufficient to develop a consistent result as the variation in results was not high between the 100- and 10,000-run simulations. Since data from the 10,000-run simulations were available, these data were used in the analysis.

The model and simulation were validated using predictive validation [48] in three ways as described in the sections below.

### 2.4.1. Model Validation

The underlying FBS model is widely accepted in the systems engineering community. The model has been through empirical studies to confirm that it is a valid model of engineering processes as shown in [17,49,50] and discussed in Section 2.1.2.

### 2.4.2. Model Parameter Validation

Model parameters were based on data that trace to real-world views from engineers. The model parameters were developed as described in Section 2.3.4. The calibration effort was shown to correlate well with expectations, showing that the calibrated FBS model behaved as expected when calibrated with the real-world data.

### 2.4.3. Validation of Simulation Output against Case Studies

Finally, published literature on the benefits of using Agile scrum over waterfall processes for software development were used to determine if the simulation output was as expected. In the literature, case studies of projects using Agile compare productivity to that when using traditional waterfall practices. These studies tend to define productivity as the amount of work that an individual can accomplish in an hour. Productivity gains tend to be 36% to 50% in most cases, with outlier examples showing gains as high as 600% and other outliers showing a productivity loss [30,32,35,39]. Other metrics, with fewer case studies for support, include a 30%–70% faster time to market and an approximately 50% reduction in defects [32]. Table 1 below examines the simulation results against these metrics.

**Table 1.** Validation of simulations against published metrics.

| Metric | Expected Result | Software Program Simulation Result | Launch Vehicle Simulation Result |
|---|---|---|---|
| Productivity | 36%–50% Gain | 42% Gain | 1% Gain |
| Time to Market | 30%–70% Faster | 62% Faster | 12% Faster |
| Reduction in Defects | ~50% Fewer | 57% Fewer | 3% More |

The software program design simulation showed a productivity gain of 42% when using Agile. This is well within the expected range, and since most of the case studies cited above are for software development efforts, it shows that the simulation accurately predicts the productivity gain software development teams can see when using Agile. The time to market, or overall length of time needed to develop the design, was 62% faster when using Agile, agreeing well with the published results of 30%–70%. Finally, defects were 57% fewer using the Agile process, close to the 50%+ (no defined range in the cited case studies) expected. Overall, the software program design simulation produced differences in results between the waterfall and Agile simulations that agreed well with all the metrics cited for comparison.

The launch vehicle simulation results are outside of the expected range when compared to the metrics. This is not an unexpected result. All the available case studies to draw metrics from use software development programs as their basis. An effort comparable to a launch vehicle design was not found in any published case study on the benefits of switching to Agile. Thus, the comparisons in Table 1 are not necessarily valid for the launch vehicle design simulation. Given the positive validation results for the software design simulation, and the fact that the same methodology was used for the

launch vehicle design simulation, the results from the launch vehicle design simulation are viewed as an extrapolation of the validated software design simulation results. Future case studies showing the productivity, time to market, and defect rate change when going from waterfall to Agile processes in the development of complex systems would be required to fully validate the simulation. Until such data are available, the results should be considered an extrapolation from a validated simulation.

## 3. Discussion

The primary method used to analyze the simulation data was statistical analysis of the results of the Monte Carlo simulation. The results were tracked through three metrics: time to complete the design effort, effort hours expended completing the design effort, and time spent in rework during the design effort. Histograms are a popular way to represent Monte Carlo simulation data and were used in this research [51]. The mean values for the team performance metrics were used for comparison. The standard deviation (SD) of the team performance metrics was also calculated to examine the variability in performance of the different processes.

### 3.1. Software Development Simulation Results Analysis

The software development simulation yielded results for using a waterfall and an Agile development process. Figure 9 represents the time needed to complete the design when using a waterfall process. The average amount of time needed to complete the design was 2012 working hours, which is close to a year worth of effort assuming a 40 h work week. The standard deviation of this parameter was 337 h, representing around two months of time.

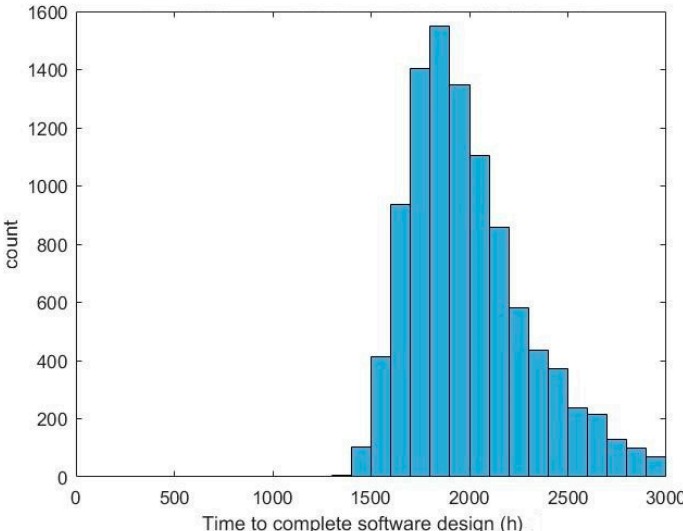

**Figure 9.** Histogram of time to complete a software program design using waterfall processes.

Figure 10 represents the time needed to complete the software program design when using an Agile process. The histogram is represented in the same scale as Figure 9 for comparison purposes. The average amount of time to complete a design was 765 h, or a little over four months with a 40 h workweek. The standard deviation was 157 h—slightly less than a month of effort. This mean is much less than that of the waterfall approach, and the standard deviation is also slightly less than half that of the waterfall approach.

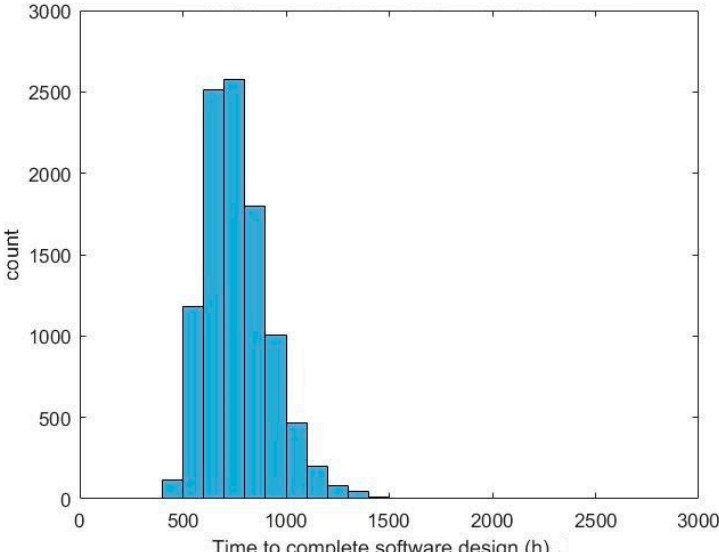

**Figure 10.** Histogram of time to complete a software program design using Agile processes.

Next, effort hours, defined as the hours of labor expended by a designer, were examined. Figure 11 shows the effort hours expended in completing the design of the software program using waterfall processes. The mean of the data was 37,038 h. The standard deviation was 1225 h. The histogram and standard deviation show that this metric was quite consistent across the 10,000 Monte Carlo runs of the simulation, with little spread in the data. This demonstrates that the total time needed to complete the design was consistent despite randomized team performance capabilities. An Anderson–Darling test of the data at a 5% significance level did not reject the null hypothesis that the data follow a normal distribution.

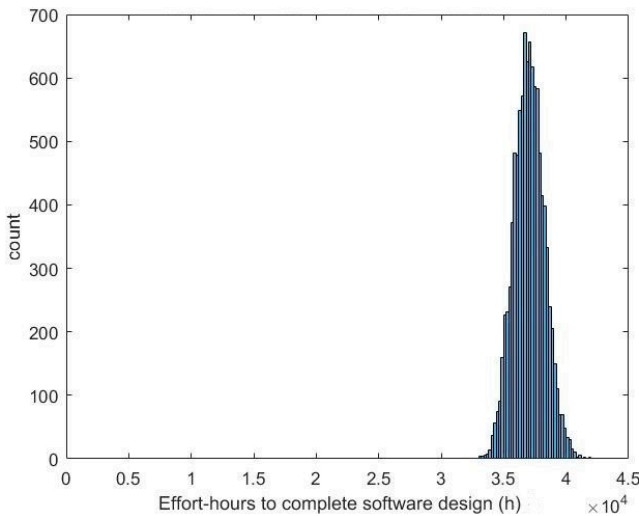

**Figure 11.** Histogram of effort hours needed to complete a software program design using waterfall processes.

Figure 12 shows the effort hours needed to complete the software program design using Agile processes. The mean of the data was 21,518 h with a standard deviation of 1091 h. Like the waterfall simulation, the Agile simulation showed a low amount of variability in the number of effort hours needed to complete the design. The mean of the Agile simulation was lower than that of the waterfall simulation by 42%. An Anderson–Darling test of the data at a 5% significance level did not reject the null hypothesis that the data follow a normal distribution.

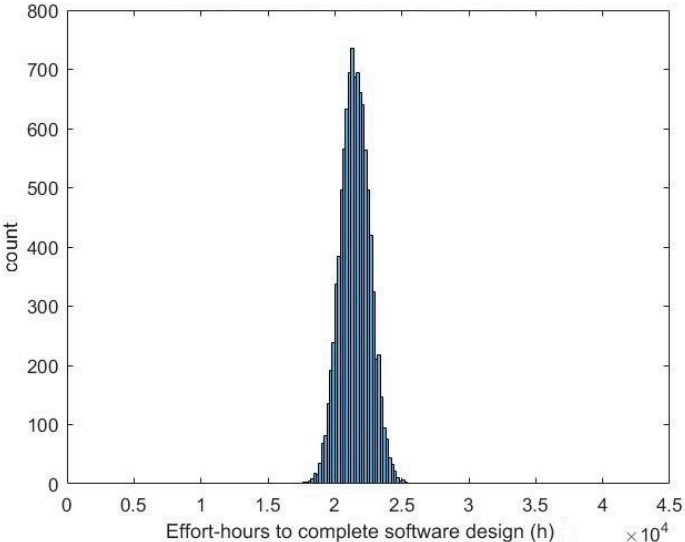

**Figure 12.** Histogram of effort hours needed to complete a software program design using Agile processes.

Rework was tracked in the simulation as the amount of time spent performing tasks that were started due to any of the three types of reformulations. Figure 13 shows the amount of rework required to complete the software program design and how it varied across the 10,000 Monte Carlo runs of the software design simulation model. The runs resulted in a mean of 1693 h spent doing rework with a 332 h standard deviation when using a waterfall process.

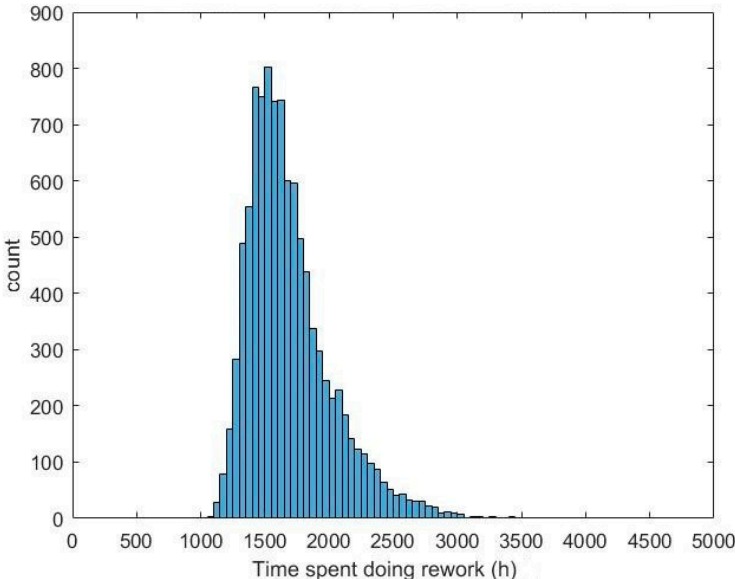

**Figure 13.** Histogram of time spent doing rework for a software program design using waterfall processes.

Figure 14 shows the amount of time spent doing rework when using an Agile process. There was a substantial reduction in time spent doing rework with the Agile process. The mean across the 10,000 simulations was 724 h and the standard deviation was 157 h. This represents a reduction in time spent on fixing defects of 57% on average.

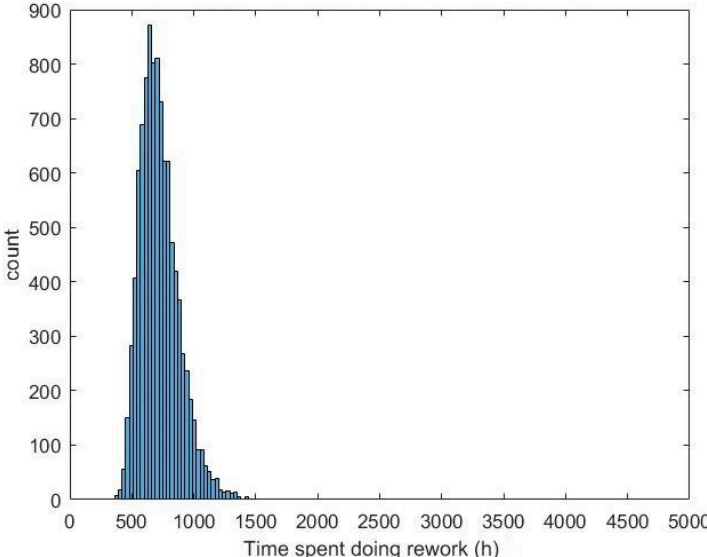

**Figure 14.** Histogram of time spent doing rework for a software program design using Agile processes.

Table 2 summarizes the results of the waterfall and Agile simulations of the software program design. The "Expected Result" column contains validation data explained in Section 2.4.3. The difference between the means of the waterfall and Agile simulation data closely matches the expected results from the dataset used to validate the simulation. The standard deviation of the effort hours expended is similar between the two simulations, showing that the amount of effort required to complete the design is fairly consistent between the waterfall and Agile processes. The standard deviation of the time to complete the design and the standard deviation of time in rework were both reduced significantly when using the Agile process. Based on the dataset used for validation, the simulation appears to correctly model the differences between the waterfall and Agile processes. The simulation also supports the claims of benefits from using the Agile process on uncoupled systems, like the uncoupled software system modeled in this simulation.

**Table 2.** Summary of software design simulation metrics.

| Metric | Waterfall Simulation | Agile Simulation | Difference in Waterfall and Agile Means | Expected Result |
|---|---|---|---|---|
| Effort hours | Mean: 37,038 h<br>SD: 1225 h | Mean: 21,518 h<br>SD: 1091 h | 42% less time in Agile mean | 36%–50% less |
| Total time expended | Mean: 2012 h<br>SD: 337 h | Mean: 765 h<br>SD: 157 h | 62% less time in Agile mean | 30%–70% less |
| Time spent in rework | Mean: 1693 h<br>SD: 332 h | Mean: 724 h<br>SD: 157 h | 57% less time in Agile mean | ~50% less |

*3.2. Launch Vehicle Development Simulation Results Analysis*

The launch vehicle development simulation produced results from 10,000 Monte Carlo runs for both waterfall and Agile processes. The time needed to complete the launch vehicle design using waterfall development processes is shown in Figure 15. The mean of this data was 1836 h, or roughly 10.5 months of time to complete the design of a simple launch vehicle. The standard deviation of the data was 405 h.

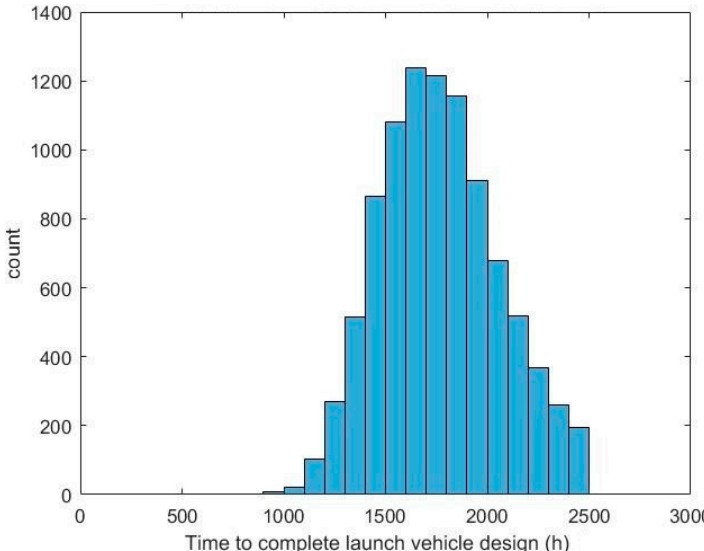

**Figure 15.** Histogram of time to complete a launch vehicle design using a waterfall process.

When using an Agile process, the time to complete the design of the simple launch vehicle had a mean of 1608 h with a standard deviation of 407 h. This is a 12% reduction in the mean when using an Agile process. The standard deviations of the two processes are very similar. A histogram of the time to complete the launch vehicle design when using the Agile process is shown in Figure 16.

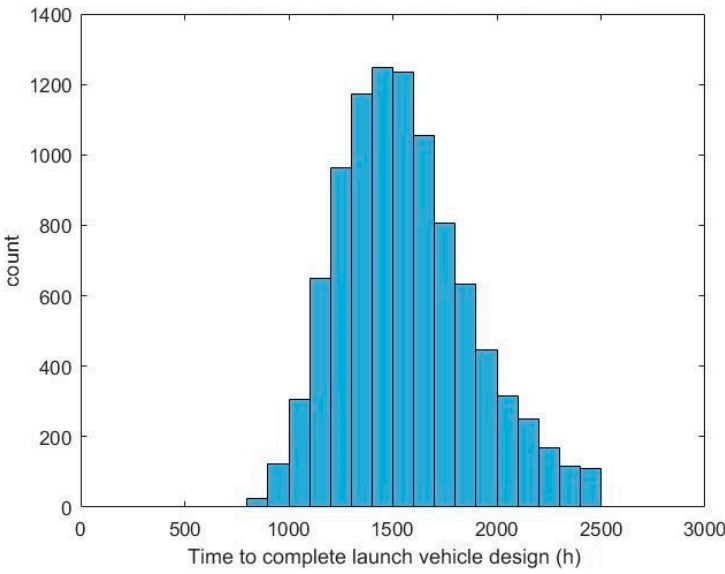

**Figure 16.** Histogram of time to complete a launch vehicle design using an Agile process.

The effort hours, or total number of hours expended by all the agents in the simulation, to complete the design when using a waterfall process across the 10,000 Monte Carlo runs is shown in Figure 17. The data had a mean of 22,846 h and a relatively low standard deviation of 1693 h. This maintains the characteristic seen in the software design simulation where the amount of effort hours needed to complete the launch vehicle design was relatively consistent between the Monte Carlo runs.

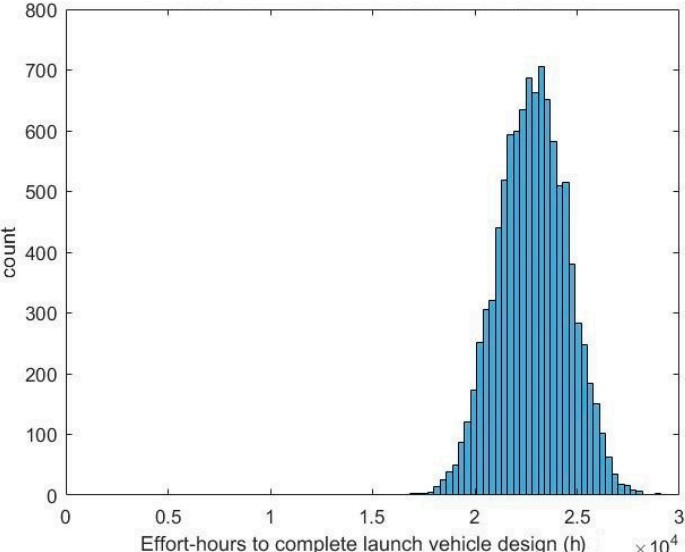

**Figure 17.** Histogram of effort hours needed to complete a launch vehicle design using a waterfall process.

Figure 18 shows a histogram of the effort hours needed to complete the simple launch vehicle design when using an Agile process. The data had a mean of 22,672 h and a standard deviation of 1724 h. These data also continued the trend seen in the software simulation, where the effort hours needed to complete the design did not show large variations across the 10,000 Monte Carlo runs. The Agile process shows a modest 1% decrease in the mean effort hours needed to complete the design when compared to the waterfall process. This represents only about 1/10 of a standard deviation, illustrating that the difference in effort is not significant.

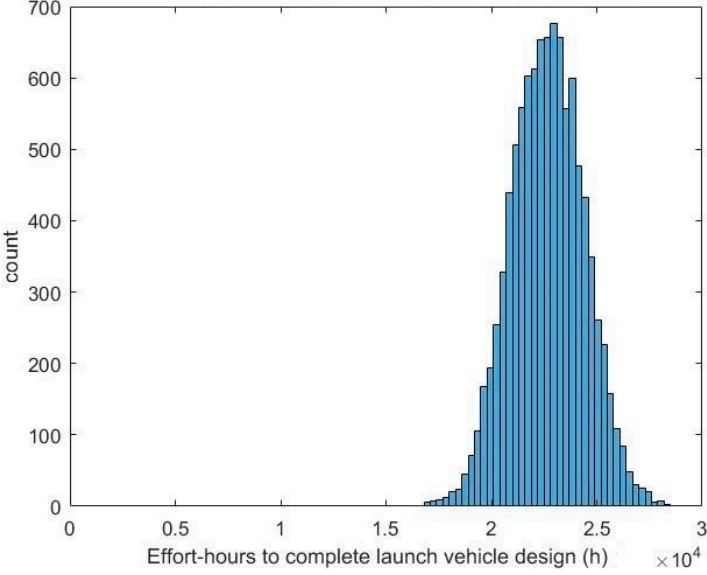

**Figure 18.** Histogram of effort hours needed to complete a launch vehicle design using an Agile process.

Rework for the launch vehicle design effort was tracked in the simulation as the effort hours expended doing design work related to type I, II, or III reformulations. It was tracked the same way as in the software program design simulation. Figure 19 shows the amount of time spent doing rework when using a waterfall process. These data had a mean of 1519 h with a standard deviation of 402 h.

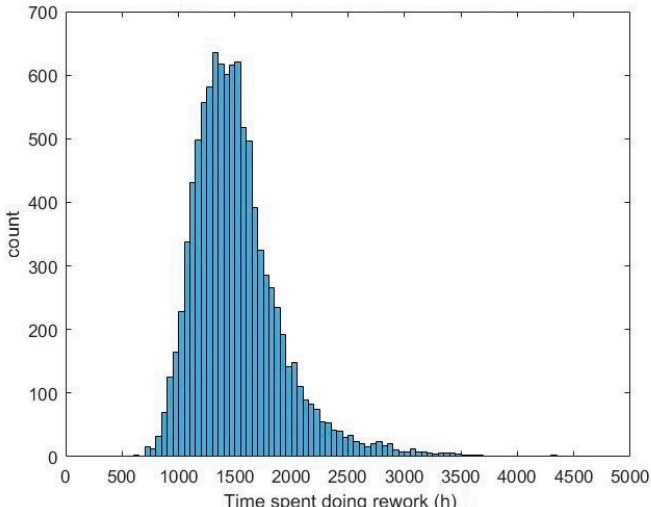

**Figure 19.** Histogram of time spent doing rework for a launch vehicle design using waterfall processes.

Figure 20 shows the amount of time spent doing rework when using an Agile process. The mean of the data was 1569 h and the standard deviation of the data was 407 h. This represents a 3% increase in the amount of effort expended on rework when using an Agile process.

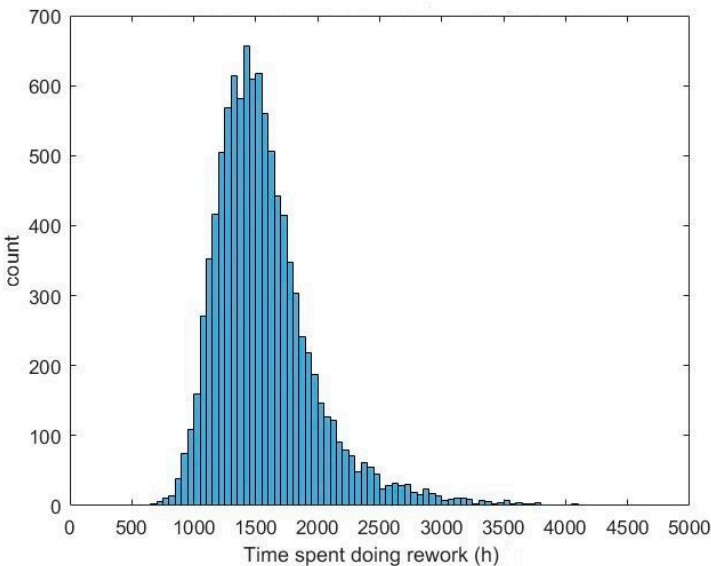

**Figure 20.** Histogram of time spent doing rework for a launch vehicle design using an Agile process.

Table 3 shows a summary of the data from the launch vehicle design simulation. The benefits seen in the software program design simulation when using Agile over waterfall processes were not as substantial when designing a highly coupled system. The effort hours and rework were very similar between the two efforts, with only a 1%–3% difference between the Agile and waterfall metrics, which, being within 1/10 of a standard deviation, is not viewed as being significant. The time taken to complete the design does show a positive benefit, but only 12% better versus 62% in the software design simulation.

The process methodology between the software and launch vehicle design simulations was the same. The primary difference between the launch vehicle and software simulations was that the launch vehicle design simulation contains significant coupling between teams, which was not present in the software design simulation. Design coupling was found to create a significant amount of rework in the design. The design coupling led to individual teams resolving their internal rework to create a

workable subsystem design that, when integrated with other parts of the launch vehicle, resulted in a deficient system design. This tended to force teams to rework a design that was functional and spawned a significant amount of effort.

**Table 3.** Summary of launch vehicle design simulation metrics.

| Metric | Waterfall Simulation | Agile Simulation | Difference in Waterfall and Agile Means |
| --- | --- | --- | --- |
| Effort-Hours | Mean: 22,846 h SD: 1693 h | Mean: 22,672 h SD: 1724 h | 1% fewer hours in Agile mean |
| Total time expended | Mean: 1836 h SD: 405 h | Mean: 1608 h SD: 407 h | 12% fewer hours in Agile mean |
| Time spent in rework | Mean: 1519 h SD: 402 h | Mean: 1569 h SD: 407 h | 3% more hours in Agile mean |

## 4. Conclusions

The results from the software simulation showed that the modeling and simulation approach matched the expected data very well, showing the validity of the simulation. This demonstrates the ability of this novel modeling and simulation approach to produce valid predictions of the difference between teams using waterfall methodology and Agile on software development projects. The launch vehicle simulation was extrapolated from this methodology to simulate a highly coupled design. It showed that for a highly coupled design, the benefits from an Agile methodology may not be as substantial as those seen on software projects. The launch vehicle simulation showed the usefulness of the simulation for studying the effects of different engineering processes. Since there is little research on the use of Agile processes for large, coupled, hardware-intensive systems such as a launch vehicle, it was not possible to fully validate the results of the launch vehicle simulation. The results of the software simulation do provide some confidence in the overall methodology, showing that using Agile methods on highly coupled systems may require special provisions to account for the coupling in the design.

**Supplementary Materials:** The source code for the simulations and output data from the simulations are available online at https://github.com/monza66/Waterfall-Agile_ABS.

**Author Contributions:** The authors provided the following contributions: conceptualization, M.B. and B.M.; methodology, software, validation, formal analysis, investigation, data curation, and writing—original draft preparation, M.B.; writing—review and editing, supervision, B.M.

**Funding:** This research received no external funding.

**Acknowledgments:** This research was indirectly supported by the generous RADM Fred Lewis scholarship provided by I/ITSEC.

**Conflicts of Interest:** The authors declare no conflict of interest.

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
