# Peer review of "Agent-Based Simulation of Hardware-Intensive Design Teams Using the Function–Behavior–Structure Framework"

_systems, doi:10.3390/systems7030037_

Round 1
Reviewer 1 Report
Waterfall process is seldom used for software development anymore in my experience - even of significant size. While this is an interesting study, maybe spiral development would have been a better choice, or a follow-on study.
A somewhat related work that might warrant citation is: COMPLEX SYSTEM SIMULATION: AGENT-BASED MODELING AND SYSTEM DYNAMICS https://pdfs.semanticscholar.org/7274/719e02853b6323720f47f4b486c47258812b.pdf
Line 104. There references might appear more consistently if they were [32,33] instead of current formatting.
Line 127. Consider replacing "already" with "existing"
Line 148. Replace "is" with "in"
Line 172. You organize your teams with 2-4 submodules. Did you do any sensitivity testing to understand what the impact would be if this was exceeded?
Figure 3 and 4. These appear somewhat fuzzy. Can you provide a higher fidelity graphic to make them easier to read?
Line 459. Link to Table 1 is corrupt
Overall, this is one of the best written and presented papers I have reviewed in the past two years across several journals.
Reviewer 2 Report
Reference missing page 13.
Reviewer 3 Report
Interesting approach is presented, A bit more extensive literature review, and the simulation data and code should be provided for reproducibility
Other minor comments
LINE NR - COMMENT
48. literature review section is very very short, should be expanded or pointers added to existing literature review
64. when FBS is first mentioned, the reference is missing (altough Gero is mentioned later in the sentence no reference is given)
FIG 1 , and other figures - source? (if the author is the source of these diagrams, ok)
93. waterfall is sequential, not concurrent activities
409. Justification for the nine total design variables, why were these chosen?
